# Comprehensive Analysis of Phosphorus-Doped Silicon Annealed by Continuous-Wave Laser Beam at High Scan Speed

**DOI:** 10.3390/ma15227886

**Published:** 2022-11-08

**Authors:** Rasheed Ayinde Taiwo, Joong-Han Shin, Yeong-Il Son

**Affiliations:** 1Department of Future Convergence Engineering, Kongju National University, Cheonan 31080, Korea; 2Department of Mechanical and Automotive Engineering, Kongju National University, Cheonan 31080, Korea

**Keywords:** laser annealing, phosphorous (P)-doped Si, electrical property, epitaxial growth, diffusion, surface roughness

## Abstract

We report an in-depth analysis of phosphorus (P)-doped silicon (Si) with a continuous-wave laser source using a high scan speed to increase the performance of semiconductor devices. We systematically characterized the P-doped Si annealed at different laser powers using four-point probe resistance measurement, transmission electron microscopy (TEM), secondary-ion mass spectroscopy, X-ray diffractometry (XRD), and atomic force microscopy (AFM). Notably, a significant reduction in sheet resistance was observed after laser annealing, which indicated the improved electrical properties of Si. TEM images confirmed the epitaxial growth of Si in an upward direction without a polycrystalline structure. Furthermore, we observed the activation of P without diffusion, irrespective of the laser power in the secondary-ion mass-spectrometry characterization. We detected negligible changes in lattice spacing for the main (400) XRD peak, showing an insignificant effect of the laser annealing on the strain. AFM images of the annealed samples in comparison with those of the as-implanted sample showed that the laser annealing did not significantly change the surface roughness. This study provides an excellent heating method with high potential to achieve an extremely low sheet resistance without diffusion of the dopant under a very high scan speed for industrial applications.

## 1. Introduction

Phosphorus (P)-doped silicon substrates (Sub-Si) are widely used in the manufacturing of advanced semiconductor materials owing to their excellent electrical properties. However, there are numerous drawbacks associated with the ion implantation process, including damage to the crystal lattice during ion bombardment and subsequent transient enhanced diffusion caused by defects during postannealing.

The P atoms become incorporated at Si substitutional sites, generating strain in Sub-Si. This strain in the semiconductor channel is responsible for the carrier mobility enhancement, which can be increased by increasing the P concentration [1,2,3,4,5]. A large P concentration on a Sub-Si is electrically inactive, which requires a laser annealing process for activation. The activation of dopants is essential to enhance the electrical properties of the fabricated material.

Although various studies have been carried out on P-doped Si to achieve a low sheet resistance by ion implantation and postannealing [6,7,8], the traditional method causes diffusion of dopants into the substrate, which weakens the electrical properties of Si [9,10,11]. Considering these problems, laser annealing is being investigated as an alternative to anneal P-doped Si with higher temperatures and simultaneously reduce the diffusion with a shorter annealing time [12,13,14]. The demand for high-performance devices and novel materials is driving the expansion of laser-based applications. For example, laser-processed thin films can be used to form local contact openings or passivate Si surfaces [15,16]. In recent studies, millisecond laser annealing has been used to minimize undesirable diffusion [17,18,19,20]. P may be activated with a low specific contact resistance [21,22,23,24] through millisecond laser annealing. There are limited reports on activation via millisecond annealing. Notably, none of those studies demonstrated the activation of P without diffusion. Therefore, it is necessary to devise an efficient method to reduce diffusion and enhance the physical and electrical properties of P-doped Si.

Considering the challenges and gaps in knowledge mentioned above, this study focused on the dopant activation of P-doped Si by millisecond laser annealing. In particular, by employing a beam scanner system, a high-scan-speed continuous-wave (CW) laser beam was used to minimize the dopant diffusion during the annealing process. A reduced sheet resistance, single-grain microstructure, absence of redistribution, well-oriented crystal structure, and almost flawless surface morphology were achieved.

A transmission electron microscopy (TEM) analysis elucidated the solidification and melting of the P-doped Si surface. It showed complete crystallization with epitaxial regrowth for amorphous Si (a-Si) with a layer thickness of approximately 80 nm. Generally, diffusion is minimized in the dopant activation process at a temperature lower than the Si melting point. On the contrary, the study provides a comprehensive analysis of the microstructural behavior of P-doped Si to attain complete crystallization without diffusion into the Sub-Si, although the process was carried out at a temperature higher than the melting point of Si. This was enabled by the high scan speed.

## 2. Methods

### 2.1. Sample Preparation and Laser Annealing Experiment

This experiment was carried out using a 200 mm wafer (produced by the Czochralski method) of p-type Si (100) as a substrate for the implantation of P ions (P+). On Sub-Si, P+ was implanted with a dose of 5 × 10^15^ atom/cm^2^ using a high-current ion implanter (Varian/Applied Materials, VIISta 80HP) at 30 keV. The implantation direction was perpendicular to the surface of the wafer. The dopant-implanted Sub-Si was then cut to 2 × 2 cm by a laser cutter after the dopant was injected.

The cut samples were subjected to preheating through a hot chuck at 400 ℃. Subsequently, laser annealing was performed on the samples at a power of 299–394 W by a CW infrared (λ=1.94 µm) laser source (IPG Photonics, TLS-1000-U) using a high scan speed (10 m/s) through a laser beam scanner system, as illustrated in Figure 1. Process conditions used in this study are shown in Table 1. The distance between the centers of the laser spots on two adjacent scan paths was fixed at 30 μm, which yielded a beam overlap of 98.5%.

### 2.2. Sample Analysis Methods

Various properties of the fabricated samples were analyzed to investigate the effect of laser irradiation on the materials by four-point probe resistance measurement, TEM, secondary-ion mass spectroscopy (SIMS), X-ray diffractometry (XRD), and atomic force microscopy (AFM).

After laser annealing, four-point probe measurements of sheet resistance were carried out for each sample using a CMT-100M system. The tips were evenly distributed with a spacing of 1 mm between them. The measurement was carried out at five different points along the annealed region. The average value was calculated to determine the most accurate sheet resistance value for each sample. A focused ion beam (FIB) milling method (FB-2100) was used to prepare the samples for the TEM analysis. After the FIB milling process, the microstructures of the as-deposited and laser annealed samples were observed via field-emission TEM using a JEM-2100F microscope operated at 200 kV. The concentration and distribution of P in Si were determined with a SIMS IMS 7f system. The strain and atomic structure in the annealed region were measured using XRD (MiniFlex 600, Rigaku, Tokyo, Japan) in the symmetric (400) direction with a Cu K_α_ radiation source (1.54 Å) in the reflection mode with a 2*θ* scan range of 3–120°. The surface roughness was measured by AFM in tapping mode to explain the surface morphology (area of 3 × 3 μm^2^), using an SPM-9700 (Shimadzu, Kyoto, Japan) system.

## 3. Results and Discussion

### 3.1. Analysis of Sheet Resistance

After laser irradiation on P-doped Si, the sheet resistance measurement was carried out on each laser-annealed sample, including the as-implanted sample, to analyze the degree of dopant activation. The sheet resistance of the as-implanted sample was measured to be 255 kΩ/sq. However, an exponential decrease in sheet resistance was observed when the samples were annealed by the laser. The decrease in sheet resistance by laser annealing has been previously [25] and recently [26] reported. These reported resistance values are similar but slightly higher than the values of this study. Similarly, a reduction in the value of sheet resistance using RTA was reported by Current et al. [27]. The lowest sheet resistance (96.7 Ω/sq) was obtained under a process temperature of 1000 ℃. As shown in Figure 2, the sheet resistance was considerably reduced from 610 to 44.8 Ω/sq as laser annealing power increased from 299 to 335 W. This indicated an enhanced electrical conductivity of the samples. Almost uniform sheet resistance was measured at 335, 353, and 394 W, with a slight reduction. The further reduction in the sheet resistance indicated a higher electrical conductivity of the samples as the laser power increased.

### 3.2. Analysis of the Structural Evolution under Different Laser Powers

The structural evolution under different laser powers was evaluated by cross-sectional TEM images of the P-doped Si, as illustrated in Figure 3. According to Figure 3a, an amorphized layer with an approximate thickness of 80 nm was formed after ion implantation. For the annealed samples, each laser power led to a similar behavior (monocrystalline) for the melt/solidification process, except for the amorphous layer formed at the top of crystalline Si (c-Si) for laser powers of 299–335 W.

At a laser power of 299 W, the crystal growth began from the surface of a-Si, as shown in Figure 3b. The laser irradiation melted the surface of a-Si, followed by an immediate high-velocity solidification from the Sub-Si. The process did not allow formation of a polycrystalline structure in the remaining liquid region. The Si regrowth occurred in an upward direction, which resulted in an epitaxial growth (Epi growth). This result is supported by the sheet resistance measurement result for this laser power, which is significantly reduced compared with the amorphous state of the as-implanted sample.

As the laser power increased from 299 to 308 W, an enhancement in crystal growth occurred, which conversely reduced the a-Si value to approximately 50 nm, as shown in Figure 3c. The two laser powers (299 and 308 W) led to Epi growth without traces of end-of-range defects. Hence, a monocrystalline structure was observed. Further Epi growth of the P-doped Si was observed for a laser power of 335 W as the a-Si value was reduced to approximately 29 nm. This signified a higher rate of dopant activation. Thus, a larger reduction in sheet resistance was achieved for a laser power of 335 W. This continuous decrease in a-Si corresponded to an improved value of sheet resistance reduction, and hence more dopants were activated.

A different behavior was observed for laser powers of 353 and 394 W, as presented in Figure 3e,f, respectively. These laser powers led to a high crystal quality upon full Epi growth of Si. It can be inferred that the complete melting of a-Si began at any slight increase in laser power higher than 335 W. The major characteristic at these laser powers was the full Epi growth with one well-defined orientation with respect to the Sub-Si seed layer under a very high scan speed. This was reflected in the values of sheet resistance measured for the two laser powers, which were almost equal, which indicated a complete melting with a full Epi growth.

Electron diffraction patterns for a-Si, Sub-Si, and Epi-Si regions are presented in Figure 4. The images reveal the characteristic structure of the selected-area electron diffraction for each stage. The red color represents the a-Si region with a diffuse form, indicating the absence of the crystalline phase. This confirms that the region consisted of the amorphous phase. The region marked by the white ring (Epi-Si) shows a regular arrangement of grains, which could be indexed to the (400) planes of the cubic Sub-Si. This confirmed the occurrence of Epi growth of the crystalline phase during the resolidification. Similarly, the diffraction pattern observed for Sub-Si represented by the yellow box was almost the same as the pattern obtained for Epi growth with a single structure. This confirmed the good crystal quality of Si (100). In general, the amount of a-Si gradually diminished with the increase in the laser power. In the melting regime, a relatively high laser power could decrease the cooling rate and result in further Epi growth of the crystalline phase during the resolidification.

### 3.3. P Concentration Profile Analysis by SIMS

The profile of P concentration was obtained by a SIMS analysis before and after laser annealing to analyze the diffusion mechanism. All P profiles were matched with the as-implanted sample, as shown in Figure 5. There is almost no redistribution of the P profile even for complete melting at high temperatures. The diffusion of dopants beyond the source and drain regions of the semiconductor device usually reduces the performance (leakage current, etc.) of the device. According to the results of the sheet resistance and P concentration profile analysis, the laser annealing technique used in this study improved the electrical properties of the device because there were no diffusion phenomena of P dopants.

### 3.4. Atomic Structure Analysis by XRD

The strain, crystallinity, and atomic structure in the annealed region in conjunction with the variation in the lattice spacing (or lattice constant) were investigated for the main XRD peak (400). Compared with the as-implanted sample peak position of 69.08° originating from the Sub-Si with a lattice spacing of 1.3586 Å, the equivalent peak position for the annealed sample for each laser power was 68.85°, 69.42°, 69.58°, and 69.10° with lattice spacings of 1.3626, 1.3528, 1.3501, and 1.3588 Å, respectively. The XRD graph in Figure 6 presents the peak shifts of all samples with the as-implanted sample as a reference. Shifts in the XRD peak are generally related to changes in the lattice spacing. A possible reason for changes in lattice spacing is the lattice distortion resulting from the rapid heating and cooling of the laser annealing process. Additionally, the P position in the lattice structure is changed by laser annealing. This change of the P position can also generate lattice distortion and the corresponding change in the lattice spacing. For a laser power of 299 W, a relatively large left shift of the (400) peak was observed. Under the laser power of 299 W, the least activation (i.e., the least taking of the lattice site by P ions) was conducted due to the least heat input. This result may have caused the different pattern (left shift) of the peak shift compared with other laser power conditions. As shown in Table 2, there was only a small variation in the lattice spacing in the range of −0.0002 to 0.009 Å, which signified that the laser annealing had a negligible effect on the strain in the annealed region. Therefore, the effect of the laser annealing on the strain was insignificant even at the highest laser power used in this study. The additional (400) peak observed at 62° was also from the Sub-Si due to the Cu K_β_ radiation source (1.39 Å). In the XRD analysis, a filter to screen out (>90%) the Cu K_β_ radiation source was used. However, even the unfiltered small amount of the Cu K_β_ radiation resulted in an additional (400) peak because the XRD signal for the monocrystalline Sub-Si was considerably strong.

### 3.5. Surface Morphology and Roughness Analysis by AFM

Figure 7 shows a two-dimensional (2D) AFM image, while Figure 8 shows the roughness measurement results for both the as-implanted and annealed samples. Notably, the 2D images of the laser-annealed samples show similar surface morphologies compared with the as-implanted sample. Furthermore, the root mean square (RMS) value of the as-implanted sample decreased after annealing at laser powers of 308, 353, and 394 W. At a laser power of 299 W, the surface roughness increased, followed by a large reduction for the remaining laser powers. The reason for the high roughness at 299 W is not clear. However, the fast cooling induced by the lowest laser power and the subsequent high solidification may have resulted in the formation of the relatively rough surface. The main characteristic of the reduction in roughness was expected to be a surface morphology with a high surface smoothness after laser annealing, despite the very high scan speed.

## 4. Conclusions

A comprehensive analysis of the P-doped Si annealed by a CW laser beam was carried out at a high scan speed. A laser-power-dependent analysis of electrical characteristics, atomic structure, dopant diffusion, crystallinity, and surface morphology was carried out. The findings of this study can be summarized as follows:
A CW laser source (λ=1.94 µm)  with a high scan speed (10 m/s) achieved through a laser beam scanner system was employed in this study. It exhibited unique characteristics in reducing the sheet resistance, which improved the electrical properties of the fabricated material.The TEM results showed a complete crystallization with Epi regrowth of an a-Si layer with a thickness of approximately 80 nm. There was no formation of a polycrystalline structure at higher temperatures during laser processing owing to the effect of the high scan speed used in this study.The P profile analysis by SIMS showed the P concentration of the fabricated material that attained complete crystallization without diffusion into the Sub-Si although the process was carried out at a temperature higher than the melting point of Si. This was enabled by the high scan speed.There was only a small variation in the lattice spacing reflected in the (400) XRD peak, which indicated that laser annealing had a negligible effect on the strain in the annealed region.The AFM analysis results of the annealed samples were almost identical to those of the as-implanted sample.


## Figures and Tables

**Figure 1 materials-15-07886-f001:**
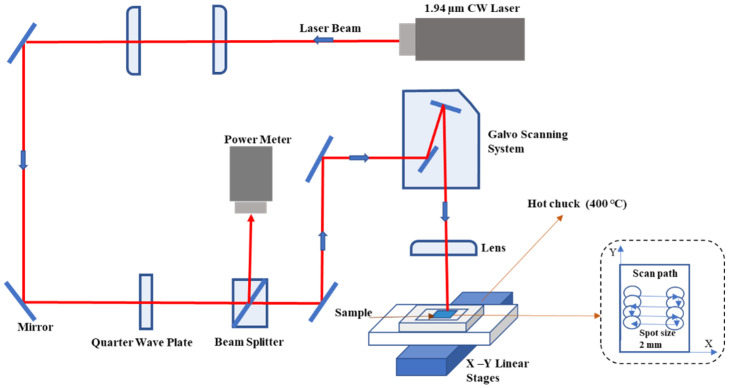
Schematic of the experimental setup for the CW laser annealing with a beam scanning system.

**Figure 2 materials-15-07886-f002:**
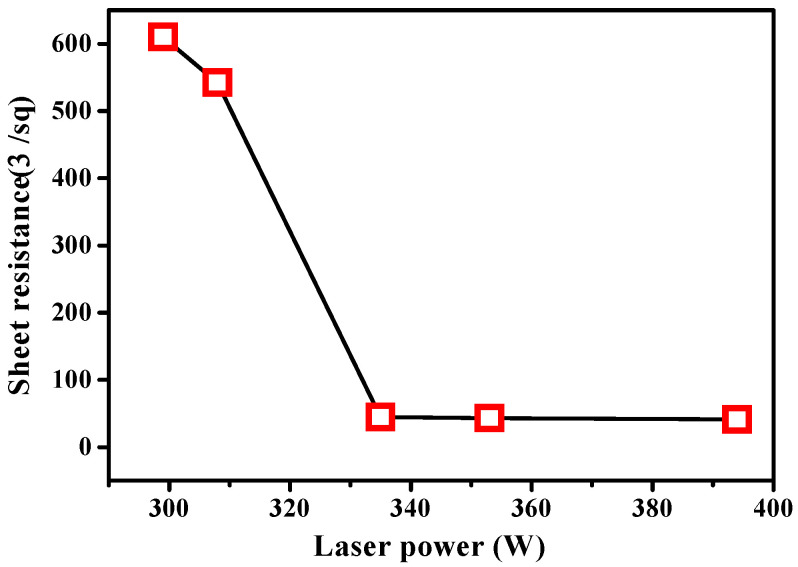
Sheet resistances of the samples annealed at different laser powers.

**Figure 3 materials-15-07886-f003:**
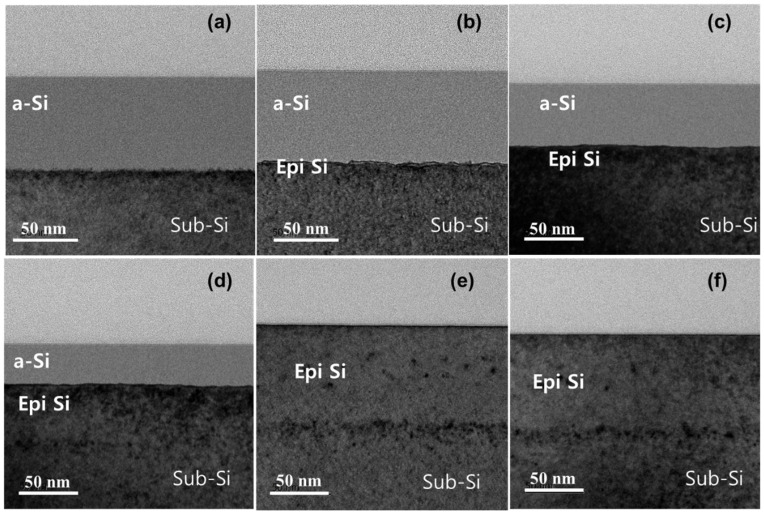
Cross-sectional TEM images of the as-implanted and annealed samples. (**a**) as-implanted sample; (**b**) 299; (**c**) 308; (**d**) 335; (**e**) 353 and (**f**) 394 W.

**Figure 4 materials-15-07886-f004:**
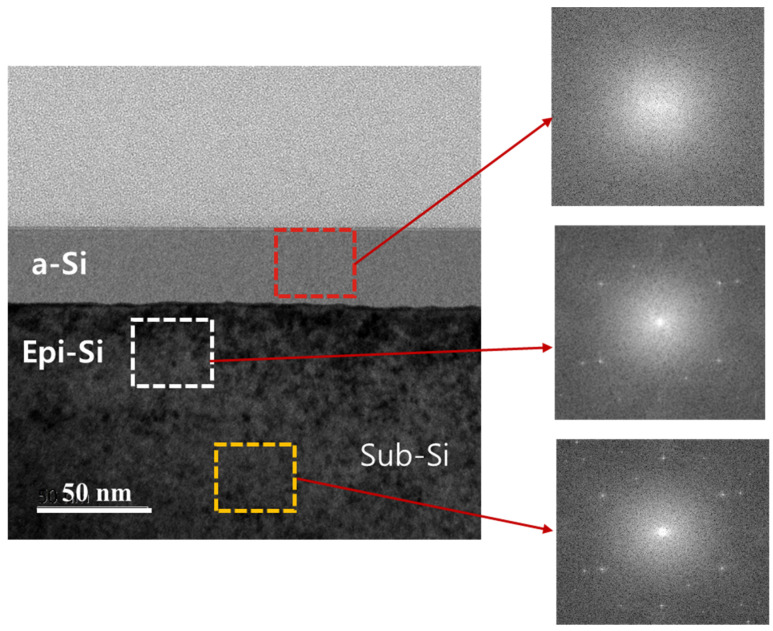
Diffraction patterns for a-Si, Sub-Si, and Epi-Si.

**Figure 5 materials-15-07886-f005:**
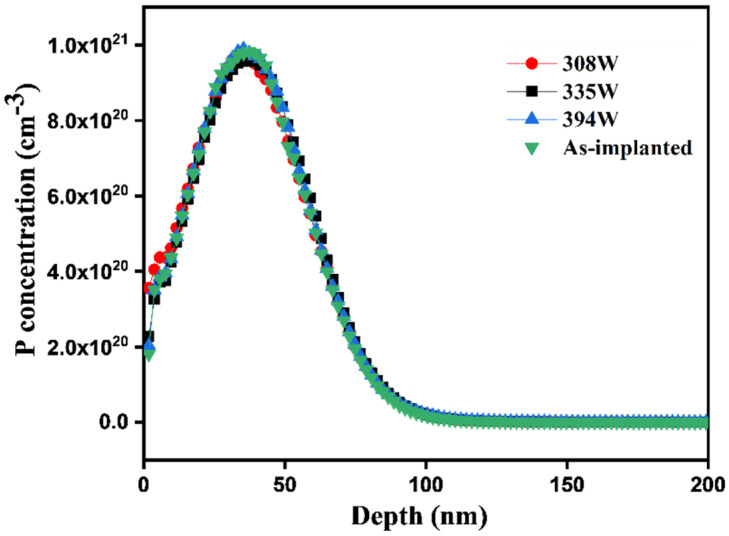
P concentration profiles of the as-implanted and annealed samples.

**Figure 6 materials-15-07886-f006:**
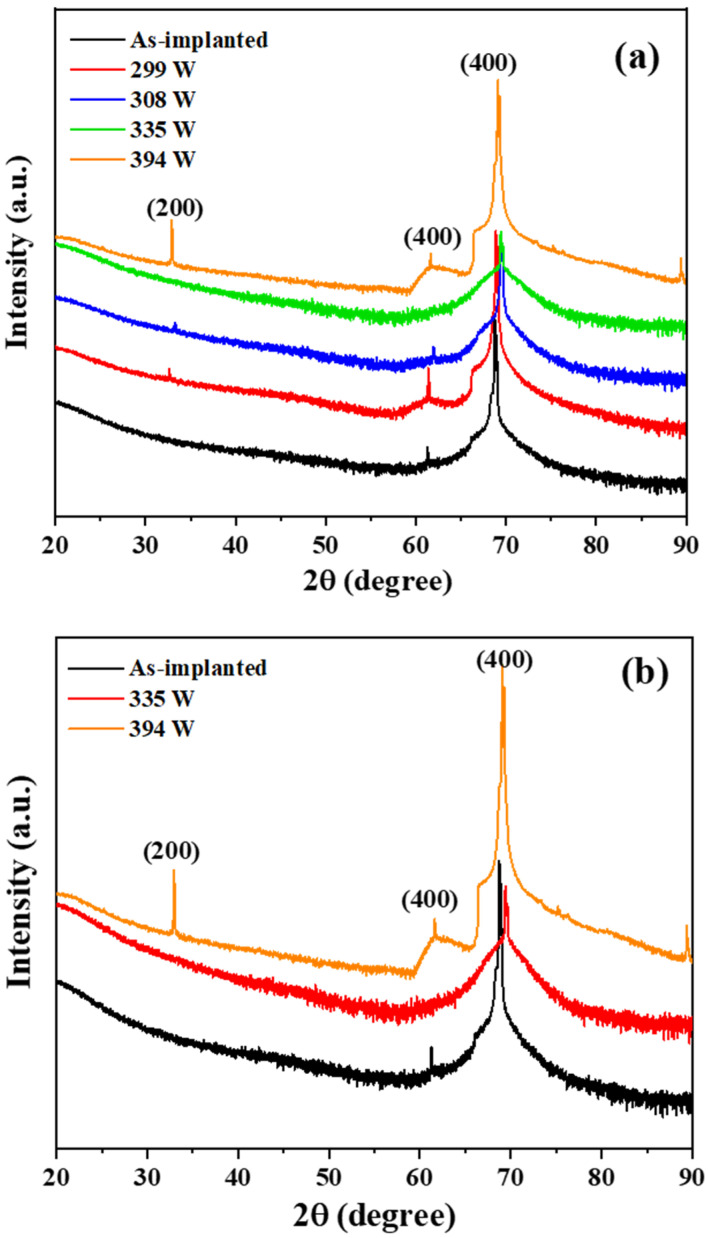
XRD analysis (*θ*/2*θ* scan). (**a**) all conditions; (**b**) as-implanted and high laser powers.

**Figure 7 materials-15-07886-f007:**
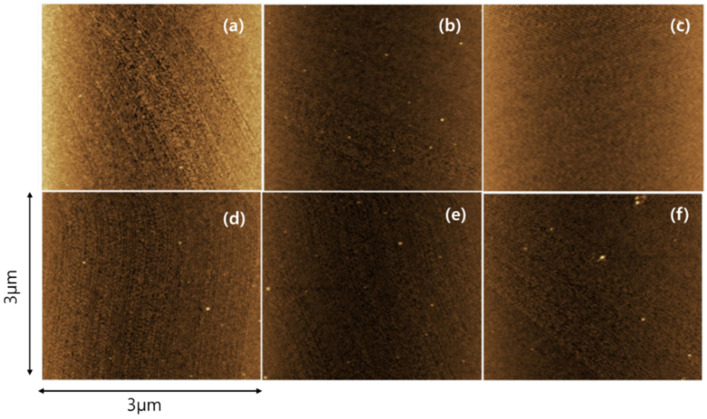
The 2D AFM images of the as-implanted and annealed samples. (**a**) as-implanted sample; (**b**) 299; (**c**) 308; (**d**) 335; (**e**) 353 and (**f**) 394 W.

**Figure 8 materials-15-07886-f008:**
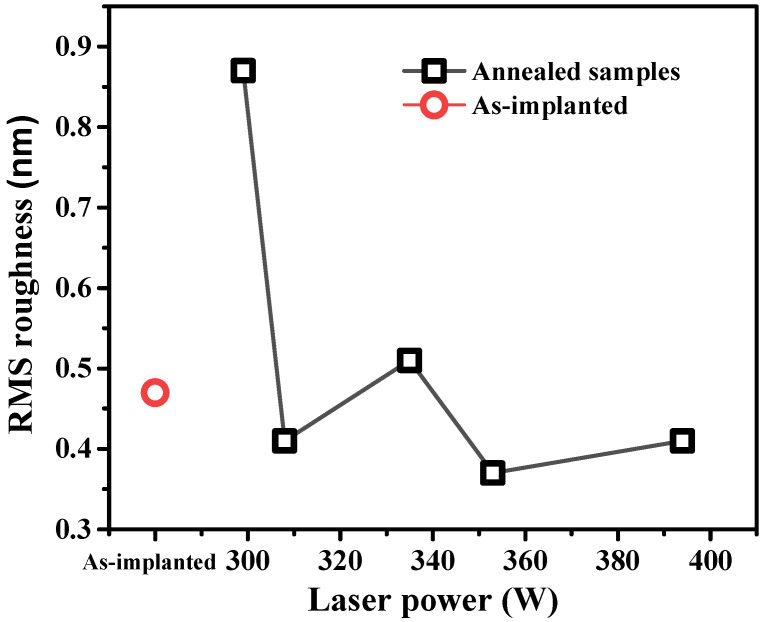
RMS roughness of the as-implanted and annealed samples.

**Table 1 materials-15-07886-t001:** Process conditions used in the laser annealing experiment.

Experiment No.	Laser Power (W)	Scan Speed (m/s)
1	299	10
2	308	10
3	335	10
4	353	10
5	394	10

**Table 2 materials-15-07886-t002:** Data for the (400) XRD peak position and corresponding lattice spacing.

Sample	Peak Position (°, 2*θ*)	Lattice Spacing (Å)	Difference in Lattice Spacing (Å)
as-implanted	69.08	1.3586	0
299	68.85	1.3626	−0.004
308	69.42	1.3528	0.006
335	69.58	1.3501	0.009
394	69.07	1.3588	−0.0002

## Data Availability

Data are available from the corresponding authors upon reasonable request.

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
