# Peer review of "Comprehensive Analysis of Phosphorus-Doped Silicon Annealed by Continuous-Wave Laser Beam at High Scan Speed"

_materials, 2022, doi:10.3390/ma15227886_

Round 1

Reviewer 1 Report

This paper reports the analysis of continuous wave (CW) laser beam activation of phosphorus ion implantation in p-type Si substrate. The dopant activation was characterized by sheet resistance. The crystalline quality and morphology of Si substrate after laser beam scanning were characterized by x-ray diffraction, TEM and AFM. A decreased sheet resistance as the result of phosphorous dopant activation by laser beam annealing was observed. The reported results are of interest for Si based device fabrication. Before the paper can be considered for publication, the authors should address following comments.

1.     In Fig. 1 the laser beam diameter is 60 um and the distance between the centers of adjacent beams is 30 um. The authors concluded the beam overlap is 97%. Please explain how 97% is calculated. The laser beam scanning speed is claimed to be 10 m/s. Please explain how to achieve this speed.

2.     Rapid thermal annealing (RTA) has been widely used in Si industry. The authors should include the results of a reference sample of phosphorous implanted Si that is annealed by RTA.

3.     In the first paragraph of section 3.1, it is written “the sheet resistance was largely reduced from 299 to 335 W”. This is not clear.

4.     Regarding the dopant activation, authors should include Hall effect measurement results to show the carrier concentration.

5.     In XRD analysis, the authors observed the shift of diffraction peaks of Si after laser beam annealing. The cause of shift should be explained. Additional peaks label with (400) are observed in samples annealed by 394W in Fig. 6 (a) and (b). The authors should explain what these peaks are and why this happens.

Author Response

To editors and reviewers:

We really appreciate your valuable comments. We are sincerely sorry for misrepresentation of some values that brought about confusion. We have revised and responded to the Reviewer's comments. The detail contents of the revision and response are described below.  

COMMENTS FOR THE AUTHOR:

Reviewer #1: This paper reports the analysis of continuous wave (CW) laser beam activation of phosphorus ion implantation in p-type Si substrate. The dopant activation was characterized by sheet resistance. The crystalline quality and morphology of Si substrate after laser beam scanning were characterized by x-ray diffraction, TEM and AFM. A decreased sheet resistance as the result of phosphorous dopant activation by laser beam annealing was observed. The reported results are of interest for Si based device fabrication. Before the paper can be considered for publication, the authors should address following comments.

1. In Fig. 1 the laser beam diameter is 60 um and the distance between the centers of adjacent beams is 30 um. The authors concluded the beam overlap is 97%. Please explain how 97% is calculated. The laser beam scanning speed is claimed to be 10 m/s. Please explain how to achieve this speed.

Ans) We thank the reviewer for pointing out our mistakes in presenting information in the text and figure. The laser beam diameter used in the experiment was not 60 um but 2 mm (2000 um). Based on this, correct beam overlap ratio was 98.5% ((2000 um – 30um)/2000 um ´ 100%). Wrong information has been corrected in the text and Figure 1. Actual wavelength of the laser beam was ‘1.94 um’ not ‘2 um’. We also have modified this and included the company and model name for the laser source. Similarly, table 1 (typo in laser power) and Figure 5 (legend) have been modified to correct the mistake (please see Page 2, line 76 and 80; Page 3, line 82 and 85; Page 7, line 183, and Page 10, line 237 for corrections).

2. Rapid thermal annealing (RTA) has been widely used in Si industry. The authors should include the results of a reference sample of phosphorous implanted Si that is annealed by RTA.

Ans) The results of phosphorous implanted Si that is annealed by RTA has been reported with appropriate reference. (Page 4, line 113 ̶116 and page 12, line 339 ̶341).       

3. In the first paragraph of section 3.1, it is written “the sheet resistance was largely reduced from 299 to 335 W”. This is not clear.

Ans) The sentence mentioned above has been modified to improve the clarity (Page 4, line 116 ̶117).    

4. Regarding the dopant activation, authors should include Hall effect measurement results to show the carrier concentration.

Ans) The reviewer is appreciated for this suggestion; however, we humbly report that this study did not include hall effect measurement as requested by the reviewer due to unavailability of the equipment as at the time of this research work. We think it could be very difficult to include it at this time. We would consider Hall effect measurement in our subsequent research.

5. In XRD analysis, the authors observed the shift of diffraction peaks of Si after laser beam annealing. The cause of shift should be explained. Additional peaks label with (400) are observed in samples annealed by 394W in Fig. 6 (a) and (b). The authors should explain what these peaks are and why this happens.

Ans) The shift of the XRD peak is generally related to the change of the lattice spacing. A possible reason for the change of the lattice spacing is the lattice distortion resulting from the rapid heating and cooling of the laser annealing process. Besides, the P position in the lattice structure is changed by laser annealing. This change of the P position also can generate the lattice distortion and the corresponding change of the lattice spacing. The additional (400) peak observed at 62° was also from a Si substrate due to a Cu-K_beta radiation source (1.39 Angstrom). In the XRD analysis, the filter to screen out (> 9+0%) the Cu-K_beta radiation source was used. However, even the unfiltered small amount of the Cu-K_beta radiation resulted in the additional (400) peak because the XRD signal for the monocrystalline Si substrate was considerably strong. We added new discussions for comment No. 5 to the manuscript. (Page 7, line 194 ̶203, line 207 ̶ 211).

Reviewer 2 Report

The manuscript presents an experimental investigation of activation of P-doped silicon by laser annealing.

The authors combine structural and electrical characterization showing that laser-annealed sample at optimized power present significantly lowered sheet resistance consisting with P activation even at very high P concentration.

I believe the subject is of interest for the audience of Materials. I however encourage the authors to revise/better explain or comment the following points, since some of their experimental findings are not discussed:

-XRD: in some spectra (394W, 299W) there are some step-like features appearing before the (400) reflections, what is their origin?

-XRD: the authors should try to discuss the origin of the different left/right shifts observed

- AFM: why is the surface roughness increased at 299W, could it be correlated to the different shift observed by XRD on this sample?

-SIMS: in the conclusions, the authors say " The P profile analysis by SIMS showed the dopant activation". This is not correct since SIMS is not directly probing P electrical activation but only the P density.

Overall, I encourage the author to expand the discussion of the experimental results.

Author Response

To editors and reviewers:

We really appreciate your valuable comments. We are sincerely sorry for misrepresentation of some values that brought about confusion. We have revised and responded to the Reviewer's comments. The detail contents of the revision and response are described below.  

Reviewer #2: The manuscript presents an experimental investigation of activation of P-doped silicon by laser annealing. The authors combine structural and electrical characterization showing that laser-annealed sample at optimized power present significantly lowered sheet resistance consisting with P activation even at very high P concentration. I believe the subject is of interest for the audience of Materials. I however encourage the authors to revise/better explain or comment the following points, since some of their experimental findings are not discussed:

-XRD: in some spectra (394 W, 299 W) there are some step-like features appearing before the (400) reflections, what is their origin?

Ans) The step-like shape appeared before the main peak (400) can be connected to the log scale applied to the y-axis. We intentionally changed the y-axis scale to log scale to easily observe the behaviour of all XRD peaks on one graph. The concerned peaks with this shape have a very high intensity compared to others.

-XRD: the authors should try to discuss the origin of the different left/right shifts observed

Ans) More discussion on the peak shift has been added to the manuscript (Page 7, line 194 ̶203).

- AFM: why is the surface roughness increased at 299 W, could it be correlated to the different shift observed by XRD on this sample?

Ans) As far as we are concerned, we could not find the reasonable correlation between the surface roughness and XRD data. The reason for the high roughness at 299 W is not clear. However, the fast cooling induced by the lowest laser power and the subsequent high solidification may result in the formation of the relatively rough surface. (Page 9, line 222 ̶ 224).

-SIMS: in the conclusions, the authors say " The P profile analysis by SIMS showed the dopant activation". This is not correct since SIMS is not directly probing P electrical activation but only the P density.

Ans) This has been corrected in the manuscript. (Page 10, line 245).

Overall, I encourage the author to expand the discussion of the experimental results.

This manuscript has been expanded to include new discussions as professionally suggested by the reviewer, modified figures, and reference number throughout. (Page 2, line 76 and 80; Page 3, line 82 and 85; Page 4, line 113 ̶116 and 116 ̶117; Page 7, line 183; Page 7, line 194 ̶203, line 207 ̶ 211; Page 9, line 222 ̶ 224; Page 10, line 237 and 245; Page 12, line 339 ̶341).

Round 2

Reviewer 1 Report

Thank you for the response and revisions in the manuscript. The quality of manuscript is improved.

Reviewer 2 Report

The authors thoroughly addressed the comments made on the original version. The paper can now be accepted for publication.